# Learning Neural Network Policies with Guided Policy Search under Unknown Dynamics

**Sergey Levine and Pieter Abbeel**
Department of Electrical Engineering and Computer Science
University of California, Berkeley
Berkeley, CA 94709
{svlevine, pabbeel}@eecs.berkeley.edu

## Abstract

We present a policy search method that uses iteratively refitted local linear models to optimize trajectory distributions for large, continuous problems. These trajectory distributions can be used within the framework of guided policy search to learn policies with an arbitrary parameterization. Our method fits time-varying linear dynamics models to speed up learning, but does not rely on learning a global model, which can be difficult when the dynamics are complex and discontinuous. We show that this hybrid approach requires many fewer samples than model-free methods, and can handle complex, nonsmooth dynamics that can pose a challenge for model-based techniques. We present experiments showing that our method can be used to learn complex neural network policies that successfully execute simulated robotic manipulation tasks in partially observed environments with numerous contact discontinuities and underactuation.

## 1 Introduction

Policy search methods can be divided into model-based algorithms, which use a model of the system dynamics, and model-free techniques, which rely only on real-world experience without learning a model [10]. Although model-free methods avoid the need to model system dynamics, they typically require policies with carefully designed, low-dimensional parameterizations [4]. On the other hand, model-based methods require the ability to learn an accurate model of the dynamics, which can be very difficult for complex systems, especially when the algorithm imposes restrictions on the dynamics representation to make the policy search efficient and numerically stable [5].

In this paper, we present a hybrid method that fits local, time-varying linear dynamics models, which are not accurate enough for standard model-based policy search. However, we can use these local linear models to efficiently optimize a time-varying linear-Gaussian controller, which induces an approximately Gaussian distribution over trajectories. The key to this procedure is to restrict the change in the trajectory distribution at each iteration, so that the time-varying linear model remains valid under the new distribution. Since the trajectory distribution is approximately Gaussian, this can be done efficiently, in terms of both sample count and computation time.

To then learn general parameterized policies, we combine this trajectory optimization method with guided policy search. Guided policy search optimizes policies by using trajectory optimization in an iterative fashion, with the policy optimized to match the trajectory, and the trajectory optimized to minimize cost and match the policy. Previous guided policy search methods used model-based trajectory optimization algorithms that required known, differentiable system dynamics [12, 13, 14]. Using our algorithm, guided policy search can be performed under unknown dynamics.

This hybrid guided policy search method has several appealing properties. First, the parameterized policy never needs to be executed on the real system – all system interaction during training is done

using the time-varying linear-Gaussian controllers. Stabilizing linear-Gaussian controllers is easier than stabilizing arbitrary policies, and this property can be a notable safety benefit when the initial parameterized policy is unstable. Second, although our algorithm relies on fitting a time-varying linear dynamics model, we show that it can handle contact-rich tasks where the true dynamics are not only nonlinear, but even discontinuous. This is because the learned linear models average the dynamics from both sides of a discontinuity in proportion to how often each side is visited, unlike standard linearization methods that differentiate the dynamics. This effectively transforms a discontinuous deterministic problem into a smooth stochastic one. Third, our algorithm can learn policies for partially observed tasks by training a parameterized policy that is only allowed to observe some parts of the state space, using a fully observed formulation for the trajectory optimizer. This corresponds to full state observation during training (for example in an instrumented environment), but only partial observation at test time, making policy search for partially observed tasks significantly easier. In our evaluation, we demonstrate this capability by training a policy for inserting a peg into hole when the precise position of the hole is unknown at test time. The learned policy, represented by a neural network, acquires a strategy that searches for and finds the hole regardless of its position.

The main contribution of our work is an algorithm for optimizing trajectories under unknown dynamics. We show that this algorithm outperforms prior methods in terms of both sample complexity and the quality of the learned trajectories. We also show that our method can be integrated with guided policy search, which previously required known models, to learn policies with an arbitrary parameterization, and again demonstrate that the resulting policy search method outperforms prior methods that optimize the parameterized policy directly. Our experimental evaluation includes simulated peg-in-hole insertion, high-dimensional octopus arm control, swimming, and bipedal walking.

## 2  Preliminaries

Policy search consists of optimizing the parameters $\theta$ of a policy $\pi_\theta(\mathbf{u}_t|\mathbf{x}_t)$, which is a distribution over actions $\mathbf{u}_t$ conditioned on states $\mathbf{x}_t$, with respect to the expectation of a cost $\ell(\mathbf{x}_t, \mathbf{u}_t)$, denoted $E_{\pi_\theta}[\sum_{t=1}^T \ell(\mathbf{x}_t, \mathbf{u}_t)]$. The expectation is under the policy and the dynamics $p(\mathbf{x}_{t+1}|\mathbf{x}_t, \mathbf{u}_t)$, which together form a distribution over trajectories $\tau$. We will use $E_{\pi_\theta}[\ell(\tau)]$ to denote the expected cost.

Our algorithm optimizes a time-varying linear-Gaussian policy $p(\mathbf{u}_t|\mathbf{x}_t) = \mathcal{N}(\mathbf{K}_t\mathbf{x}_t + \mathbf{k}_t, \mathbf{C}_t)$, which allows for a particularly efficient optimization method when the initial state distribution is narrow and approximately Gaussian. Arbitrary parameterized policies $\pi_\theta$ are optimized using the guided policy search technique, in which $\pi_\theta$ is trained to match one or more Gaussian policies $p$. In this way, we can learn a policy that succeeds from many initial states by training a single stationary, nonlinear policy $\pi_\theta$, which might be represented (for example) by a neural network, from multiple Gaussian policies. As we show in Section 5, this approach can outperform methods that search for the policy parameters $\theta$ directly, by taking advantage of the linear-Gaussian structure of $p$ to accelerate learning. For clarity, we will refer to $p$ as a trajectory distribution since, for a narrow $\mathbf{C}_t$ and well-behaved dynamics, it induces an approximately Gaussian distribution over trajectories, while the term "policy" will be reserved for the parameterized policy $\pi_\theta$.

Time-varying linear-Gaussian policies have previously been used in a number of model-based and model-free methods [25, 16, 14] due to their close connection with linear feedback controllers, which are frequently used in classic deterministic trajectory optimization. The algorithm we will describe builds on the iterative linear-Gaussian regulator (iLQG), which optimizes trajectories by iteratively constructing locally optimal linear feedback controllers under a local linearization of the dynamics and a quadratic expansion of the cost [15]. Under linear dynamics and quadratic costs, the value or cost-to-go function is quadratic, and can be computed with dynamic programming. The iLQG algorithm alternates between computing the quadratic value function around the current trajectory, and updating the trajectory using a rollout of the corresponding linear feedback controller.

We will use subscripts to denote derivatives, so that $\ell_{\mathbf{xu}t}$ is the derivative of the cost at time step $t$ with respect to $(\mathbf{x}_t, \mathbf{u}_t)^\mathrm{T}$, $\ell_{\mathbf{xu},\mathbf{xu}t}$ is the Hessian, $\ell_{\mathbf{x}t}$ is the derivative with respect to $\mathbf{x}_t$, and so forth. Using $\mathcal{N}(f_{\mathbf{x}t}\mathbf{x}_t + f_{\mathbf{u}t}\mathbf{u}_t, \mathbf{F}_t)$ to denote the local linear-Gaussian approximation to the dynamics, iLQG computes the first and second derivatives of the Q and value functions as follows:

$$Q_{\mathbf{xu},\mathbf{xu}t} = \ell_{\mathbf{xu},\mathbf{xu}t} + f_{\mathbf{xu}t}^\mathrm{T} V_{\mathbf{x},\mathbf{x}t+1} f_{\mathbf{xu}t} \qquad Q_{\mathbf{xu}t} = \ell_{\mathbf{xu}t} + f_{\mathbf{xu}t}^\mathrm{T} V_{\mathbf{x}t+1} \qquad (1)$$
$$V_{\mathbf{x},\mathbf{x}t} = Q_{\mathbf{x},\mathbf{x}t} - Q_{\mathbf{u},\mathbf{x}t}^\mathrm{T} Q_{\mathbf{u},\mathbf{u}t}^{-1} Q_{\mathbf{u},\mathbf{x}} \qquad\qquad V_{\mathbf{x}t} = Q_{\mathbf{x}t} - Q_{\mathbf{u},\mathbf{x}t}^\mathrm{T} Q_{\mathbf{u},\mathbf{u}t}^{-1} Q_{\mathbf{u}t}$$

The linear controller $g(\mathbf{x}_t) = \hat{\mathbf{u}}_t + \mathbf{k}_t + \mathbf{K}_t(\mathbf{x}_t - \hat{\mathbf{x}}_t)$ can be shown to minimize this quadratic Q-function, where $\hat{\mathbf{x}}_t$ and $\hat{\mathbf{u}}_t$ are the states and actions of the current trajectory, $\mathbf{K}_t = -Q_{\mathbf{u},\mathbf{u}t}^{-1}Q_{\mathbf{u},\mathbf{x}t}$, and $\mathbf{k}_t = -Q_{\mathbf{u},\mathbf{u}t}^{-1}Q_{\mathbf{u}t}$. We can also construct a linear-Gaussian controller with the mean given by the deterministic optimal solution, and the covariance proportional to the curvature of the Q-function:

$$p(\mathbf{u}_t|\mathbf{x}_t) = \mathcal{N}(\bar{\mathbf{u}}_t + \mathbf{k}_t + \mathbf{K}_t(\mathbf{x}_t - \hat{\mathbf{x}}_t), Q_{\mathbf{u},\mathbf{u}t}^{-1})$$

Prior work has shown that this distribution optimizes a maximum entropy objective [12], given by

$$p(\tau) = \arg \min_{p(\tau) \in \mathcal{N}(\tau)} E_p[\ell(\tau)] - \mathcal{H}(p(\tau)) \text{ s.t. } p(\mathbf{x}_{t+1}|\mathbf{x}_t, \mathbf{u}_t) = \mathcal{N}(\mathbf{x}_{t+1}; f_{\mathbf{x}t}\mathbf{x}_t + f_{\mathbf{u}t}\mathbf{u}_t, \mathbf{F}_t), \quad (2)$$

where $\mathcal{H}$ is the differential entropy. This means that the linear-Gaussian controller produces the widest, highest-entropy distribution that also minimizes the expected cost, subject to the linearized dynamics and quadratic cost function. Although this objective differs from the expected cost, it is useful as an intermediate step in algorithms that optimizes the more standard expected cost objective [20, 12]. Our method similarly uses the maximum entropy objective as an intermediate step, and converges to trajectory distribution with the optimal expected cost. However, unlike iLQG, our method operates on systems where the dynamics are unknown.

## 3   Trajectory Optimization under Unknown Dynamics

When the dynamics $\mathcal{N}(f_{\mathbf{x}t}\mathbf{x}_t + f_{\mathbf{u}t}\mathbf{u}_t, \mathbf{F}_t)$ are unknown, we can estimate them using samples $\{(\mathbf{x}_{ti}, \mathbf{u}_{ti})^{\mathrm{T}}, \mathbf{x}_{t+1i}\}$ from the real system under the previous linear-Gaussian controller $p(\mathbf{u}_t|\mathbf{x}_t)$, where $\tau_i = \{\mathbf{x}_{1i}, \mathbf{u}_{1i}, \ldots, \mathbf{x}_{Ti}, \mathbf{u}_{Ti}\}$ is the $i^{\mathrm{th}}$ rollout. Once we estimate the linear-Gaussian dynamics at each time step, we can simply run the dynamic programming algorithm in the preceding section to obtain a new linear-Gaussian controller. However, the fitted dynamics are only valid in a local region around the samples, while the new controller generated by iLQG can be arbitrarily different from the old one. The fully model-based iLQG method addresses this issue with a line search [23], which is impractical when the rollouts must be stochastically sampled from the real system. Without the line search, large changes in the trajectory will cause the algorithm to quickly fall into unstable, costly parts of the state space, preventing convergence. We address this issue by limiting the change in the trajectory distribution in each dynamic programming pass by imposing a constraint on the KL-divergence between the old and new trajectory distribution.

### 3.1   KL-Divergence Constraints

Under linear-Gaussian controllers, a KL-divergence constraint against the previous trajectory distribution $\hat{p}(\tau)$ can be enforced with a simple modification of the cost function. Omitting the dynamics constraint for clarity, the constrained problem is given by

$$\min_{p(\tau) \in \mathcal{N}(\tau)} E_p[\ell(\tau)] \text{ s.t. } D_{\mathrm{KL}}(p(\tau)\|\hat{p}(\tau)) \leq \epsilon.$$

This type of policy update has previously been proposed by several authors in the context of policy search [1, 19, 17]. The objective of this optimization is the standard expected cost objective, and solving this problem repeatedly, each time setting $\hat{p}(\tau)$ to the last $p(\tau)$, will minimize $E_{p(\mathbf{x}_t, \mathbf{u}_t)}[\ell(\mathbf{x}_t, \mathbf{u}_t)]$. Using $\eta$ to represent the dual variable, the Lagrangian of this problem is

$$\mathcal{L}_{\mathrm{traj}}(p(\tau), \eta) = E_p[\ell(\tau)] + \eta[D_{\mathrm{KL}}(p(\tau)\|\hat{p}(\tau)) - \epsilon].$$

Since $p(\mathbf{x}_{t+1}|\mathbf{x}_t, \mathbf{u}_t) = \hat{p}(\mathbf{x}_{t+1}|, \mathbf{x}_t, \mathbf{u}_t) = \mathcal{N}(f_{\mathbf{x}t}\mathbf{x}_t + f_{\mathbf{u}t}\mathbf{u}_t, \mathbf{F}_t)$ due to the linear-Gaussian dynamics assumption, the Lagrangian can be rewritten as

$$\mathcal{L}_{\mathrm{traj}}(p(\tau), \eta) = \left[\sum_t E_{p(\mathbf{x}_t, \mathbf{u}_t)}[\ell(\mathbf{x}_t, \mathbf{u}_t) - \eta \log \hat{p}(\mathbf{u}_t|\mathbf{x}_t)]\right] - \eta\mathcal{H}(p(\tau)) - \eta\epsilon.$$

Dividing both sides of this equation by $\eta$ gives us an objective of the same form as Equation (2), which means that under linear dynamics we can minimize the Lagrangian with respect to $p(\tau)$ using the dynamic programming algorithm from the preceding section, with an augmented cost function $\tilde{\ell}(\mathbf{x}_t, \mathbf{u}_t) = \frac{1}{\eta}\ell(\mathbf{x}_t, \mathbf{u}_t) - \log \hat{p}(\mathbf{u}_t|\mathbf{x}_t)$. We can therefore solve the original constrained problem by using dual gradient descent [2], alternating between using dynamic programming to

minimize the Lagrangian with respect to $p(\tau)$, and adjust the dual variable according to the amount of constraint violation. Using a bracket linesearch with quadratic interpolation [7], this procedure usually converges within a few iterations, especially if we accept approximate constraint satisfaction, for example by stopping when the KL-divergence is within $10\%$ of $\epsilon$. Empirically, we found that the line search tends to require fewer iterations in log space, treating the dual as a function of $\nu = \log \eta$, which also has the convenient effect of enforcing the positivity of $\eta$.

The dynamic programming pass does not guarantee that $Q_{\mathbf{u},\mathbf{u}t}^{-1}$, which is the covariance of the linear-Gaussian controller, will always remain positive definite, since nonconvex cost functions can introduce negative eigenvalues into Equation (1) [23]. To address this issue, we can simply increase $\eta$ until each $Q_{\mathbf{u},\mathbf{u}t}$ becomes positive definite, which is always possible, since the positive definite precision matrix of $\hat{p}(\mathbf{u}_t|\mathbf{x}_t)$, multiplied by $\eta$, enters additively into $Q_{\mathbf{u},\mathbf{u}t}$. This might sometimes result in the KL-divergence being lower than $\epsilon$, though this happens rarely in practice.

The step $\epsilon$ can be adaptively adjusted based on the discrepancy between the improvement in total cost predicted under the linear dynamics and quadratic cost approximation, and the actual improvement, which can be estimated using the new linear dynamics and quadratic cost. Since these quantities only involve expectations of quadratics under Gaussians, they can be computed analytically.

The amount of improvement obtained from optimizing $p(\tau)$ depends on the accuracy of the estimated dynamics. In general, the sample complexity of this estimation depends on the dimensionality of the state. However, the dynamics at nearby time steps and even successive iterations are correlated, and we can exploit this correlation to reduce the required number of samples.

## 3.2   Background Dynamics Distribution

When fitting the dynamics, we can use priors to greatly reduce the number of samples required at each iteration. While these priors can be constructed using domain knowledge, a more general approach is to construct the prior from samples at other time steps and iterations, by fitting a background dynamics distribution as a kind of crude global model. For physical systems such as robots, a good choice for this distribution is a Gaussian mixture model (GMM), which corresponds to softly piecewise linear dynamics. The dynamics of a robot can be reasonably approximated with such piecewise linear functions [9], and they are well suited for contacts, which are approximately piecewise linear with a hard boundary. If we build a GMM over vectors $(\mathbf{x}_t, \mathbf{u}_t, \mathbf{x}_{t+1})^{\mathrm{T}}$, we see that within each cluster $c_i$, the conditional $c_i(\mathbf{x}_{t+1}|\mathbf{x}_t, \mathbf{u}_t)$ represents a linear-Gaussian dynamics model, while the marginal $c_i(\mathbf{x}_t, \mathbf{u}_t)$ specifies the region of the state-action space where this model is valid.

Although the GMM models (softly) piecewise linear dynamics, it is not necessarily a good forward model, since the marginals $c_i(\mathbf{x}_t, \mathbf{u}_t)$ will not always delineate the correct boundary between two linear modes. In the case of contacts, the boundary might have a complex shape that is not well modeled by a GMM. However, if we use the GMM to obtain a prior for linear regression, it is easy to determine the correct linear mode from the covariance of $(\mathbf{x}_{ti}, \mathbf{u}_{ti})$ with $\mathbf{x}_{t+1i}$ in the current samples at time step $t$. The time-varying linear dynamics can then capture different linear modes at different time steps depending on the actual observed transitions, even if the states are very similar.

To use the GMM to construct a prior for the dynamics, we refit the GMM at each iteration to all of the samples at all time steps from the current iteration, as well as several prior interations, in order to ensure that sufficient samples are available. We then estimate the time-varying linear dynamics by fitting a Gaussian to the samples $\{\mathbf{x}_{ti}, \mathbf{u}_{ti}, \mathbf{x}_{t+1i}\}$ at each time step, which can be conditioned on $(\mathbf{x}_t, \mathbf{u}_t)^{\mathrm{T}}$ to obtain linear-Gaussian dynamics. The GMM is used to produce a normal-inverse-Wishart prior for the mean and covariance of this Gaussian at each time step. To obtain the prior, we infer the cluster weights for the samples at the current time step, and then use the weighted mean and covariance of these clusters as the prior parameters. We found that the best results were produced by large mixtures that modeled the dynamics in high detail. In practice, the GMM allowed us to reduce the samples at each iteration by a factor of 4 to 8, well below the dimensionality of the system.

## 4   General Parameterized Policies

The algorithm in the preceding section optimizes time-varying linear-Gaussian controllers. To learn arbitrary parameterized policies, we combine this algorithm with a guided policy search (GPS) ap-

---

**Algorithm 1** Guided policy search with unknown dynamics

---
1: **for** iteration $k = 1$ to $K$ **do**
2:     Generate samples $\{\tau_i^j\}$ from each linear-Gaussian controller $p_i(\tau)$ by performing rollouts
3:     Fit the dynamics $p_i(\mathbf{x}_{t+1}|\mathbf{x}_t, \mathbf{u}_t)$ to the samples $\{\tau_i^j\}$
4:     Minimize $\sum_{i,t} \lambda_{i,t} D_{\text{KL}}(p_i(\mathbf{x}_t)\pi_\theta(\mathbf{u}_t|\mathbf{x}_t)\|p_i(\mathbf{x}_t, \mathbf{u}_t))$ with respect to $\theta$ using samples $\{\tau_i^j\}$
5:     Update $p_i(\mathbf{u}_t|\mathbf{x}_t)$ using the algorithm in Section 3 and the supplementary appendix
6:     Increment dual variables $\lambda_{i,t}$ by $\alpha D_{\text{KL}}(p_i(\mathbf{x}_t)\pi_\theta(\mathbf{u}_t|\mathbf{x}_t)\|p_i(\mathbf{x}_t, \mathbf{u}_t))$
7: **end for**
8: **return** optimized policy parameters $\theta$

---

proach. In GPS methods, the parameterized policy is trained in supervised fashion to match samples from a trajectory distribution, and the trajectory distribution is optimized to minimize both its cost and difference from the current policy, thereby creating a good training set for the policy. By turning policy optimization into a supervised problem, GPS algorithms can train complex policies with thousands of parameters [12, 14], and since our trajectory optimization algorithm exploits the structure of linear-Gaussian controllers, it can optimize the individual trajectories with fewer samples than general-purpose model-free methods. As a result, the combined approach can learn complex policies that are difficult to train with prior methods, as shown in our evaluation.

We build on the recently proposed constrained GPS algorithm, which enforces agreement between the policy and trajectory by means of a soft KL-divergence constraint [14]. Constrained GPS optimizes the maximum entropy objective $E_{\pi_\theta}[\ell(\tau)] - \mathcal{H}(\pi_\theta)$, but our trajectory optimization method allows us to use the more standard expected cost objective, resulting in the following optimization:

$$\min_{\theta, p(\tau)} E_{p(\tau)}[\ell(\tau)] \text{ s.t. } D_{\text{KL}}(p(\mathbf{x}_t)\pi_\theta(\mathbf{u}_t|\mathbf{x}_t)\|p(\mathbf{x}_t, \mathbf{u}_t)) = 0 \,\forall t.$$

If the constraint is enforced exactly, the policy $\pi_\theta(\mathbf{u}_t|\mathbf{x}_t)$ is identical to $p(\mathbf{u}_t|\mathbf{x}_t)$, and the optimization minimizes the cost under $\pi_\theta$, given by $E_{\pi_\theta}[\ell(\tau)]$. Constrained GPS enforces these constraints softly, so that $\pi_\theta$ and $p$ gradually come into agreement over the course of the optimization. In general, we can use multiple distributions $p_i(\tau)$, with each trajectory starting from a different initial state or in different conditions, but we will omit the subscript for simplicity, since each $p_i(\tau)$ is treated identically and independently. The Lagrangian of this problem is given by

$$\mathcal{L}_{\text{GPS}}(\theta, p, \lambda) = E_{p(\tau)}[\ell(\tau)] + \sum_{t=1}^T \lambda_t D_{\text{KL}}(p(\mathbf{x}_t)\pi_\theta(\mathbf{u}_t|\mathbf{x}_t)\|p(\mathbf{x}_t, \mathbf{u}_t)).$$

The GPS Lagrangian is minimized with respect to $\theta$ and $p(\tau)$ in alternating fashion, with the dual variables $\lambda_t$ updated to enforce constraint satisfaction. Optimizing $\mathcal{L}_{\text{GPS}}$ with respect to $p(\tau)$ corresponds to trajectory optimization, which in our case involves dual gradient descent on $\mathcal{L}_{\text{traj}}$ in Section 3.1, and optimizing with respect $\theta$ corresponds to supervised policy optimization to minimize the weighted sum of KL-divergences. The constrained GPS method also uses dual gradient descent to update the dual variables, but we found that in practice, it is unnecessary (and, in the unknown model setting, extremely inefficient) to optimize $\mathcal{L}_{\text{GPS}}$ with respect to $p(\tau)$ and $\theta$ to convergence prior to each dual variable update. Instead, we increment the dual variables after each iteration with a multiple $\alpha$ of the KL-divergence ($\alpha = 10$ works well), which corresponds to a penalty method. Note that the dual gradient descent on $\mathcal{L}_{\text{traj}}$ during trajectory optimization is unrelated to the policy constraints, and is treated as an inner loop black-box optimizer by GPS.

Pseudocode for our modified constrained GPS method is provided in Algorithm 1. The policy KL-divergence terms in the objective also necessitate a modified dynamic programming method, which can be found in prior work [14], but the step size constraints are still enforced as described in the preceding section, by modifying the cost. The same samples that are used to fit the dynamics are also used to train the policy, with the policy trained to minimize $\lambda_t D_{\text{KL}}(\pi_\theta(\mathbf{u}_t|\mathbf{x}_{ti})\|p(\mathbf{u}_t|\mathbf{x}_{ti}))$ at each sampled state $\mathbf{x}_{ti}$. Further details about this algorithm can be found in the supplementary appendix.

Although this method optimizes the expected cost of the policy, due to the alternating optimization, its entropy tends to remain high, since both the policy and trajectory must decrease their entropy together to satisfy the constraint, which requires many alternating steps. To speed up this process, we found it useful to regularize the policy by penalizing its entropy directly, which speeds up convergence and produces more deterministic policies.

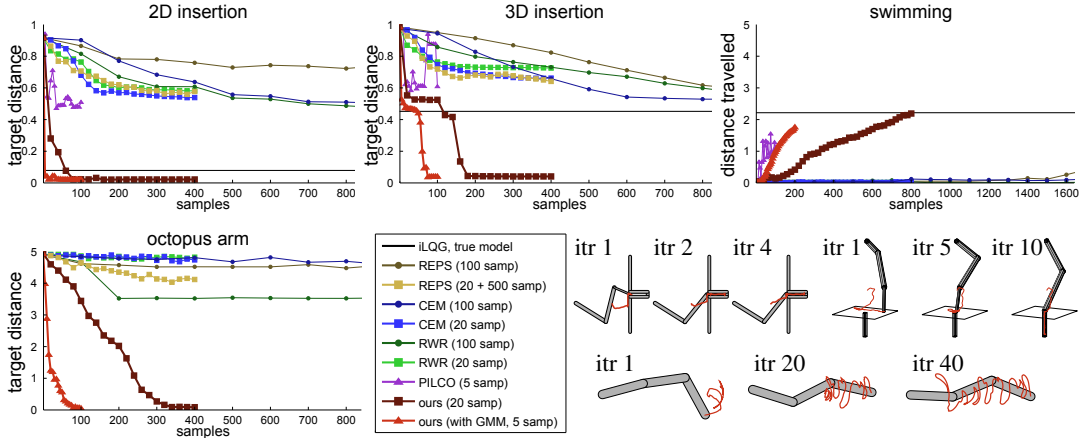

Figure 1: Results for learning linear-Gaussian controllers for 2D and 3D insertion, octopus arm, and swimming. Our approach uses fewer samples and finds better solutions than prior methods, and the GMM further reduces the required sample count. Images in the lower-right show the last time step for each system at several iterations of our method, with red lines indicating end effector trajectories.

## 5   Experimental Evaluation

We evaluated both the trajectory optimization method and general policy search on simulated robotic manipulation and locomotion tasks. The state consisted of joint angles and velocities, and the actions corresponded to joint torques. The parameterized policies were neural networks with one hidden layer and a soft rectifier nonlinearity of the form $a = \log(1 + \exp(z))$, with learned diagonal Gaussian noise added to the outputs to produce a stochastic policy. This policy class was chosen for its expressiveness, to allow the policy to learn a wide range of strategies. However, due to its high dimensionality and nonlinearity, it also presents a serious challenge for policy search methods.

The tasks are 2D and 3D peg insertion, octopus arm control, and planar swimming and walking. The insertion tasks require fitting a peg into a narrow slot, a task that comes up, for example, when inserting a key into a keyhole, or assembly with screws or nails. The difficulty stems from the need to align the peg with the slot and the complex contacts between the peg and the walls, which result in discontinuous dynamics. Control in the presence of contacts is known to be challenging, and this experiment is important for ascertaining how well our method can handle such discontinuities. Octopus arm control involves moving the tip of a flexible arm to a goal position [6]. The challenge in this task stems from its high dimensionality: the arm has 25 degrees of freedom, corresponding to 50 state dimensions. The swimming task requires controlling a three-link snake, and the walking task requires a seven-link biped to maintain a target velocity. The challenge in these tasks comes from underactuation. Details of the simulation and cost for each task are in the supplementary appendix.

### 5.1   Trajectory Optimization

Figure 1 compares our method with prior work on learning linear-Gaussian controllers for peg insertion, octopus arm, and swimming (walking is discussed in the next section). The horizontal axis shows the total number of samples, and the vertical axis shows the minimum distance between the end of the peg and the bottom of the slot, the distance to the target for the octopus arm, or the total distance travelled by the swimmer. Since the peg is $0.5$ units long, distances above this amount correspond to controllers that cannot perform an insertion.

We compare to REPS [17], reward-weighted regression (RWR) [18, 11], the cross-entropy method (CEM) [21], and PILCO [5]. We also use iLQG [15] with a known model as a baseline, shown as a black horizontal line. REPS is a model-free method that, like our approach, enforces a KL-divergence constraint between the new and old policy. We compare to a variant of REPS that also fits linear dynamics to generate 500 pseudo-samples [16], which we label "REPS (20 + 500)." RWR is an EM algorithm that fits the policy to previous samples weighted by the exponential of their reward, and CEM fits the policy to the best samples in each batch. With Gaussian trajectories, CEM and RWR only differ in the weights. These methods represent a class of RL algorithms that fit the policy

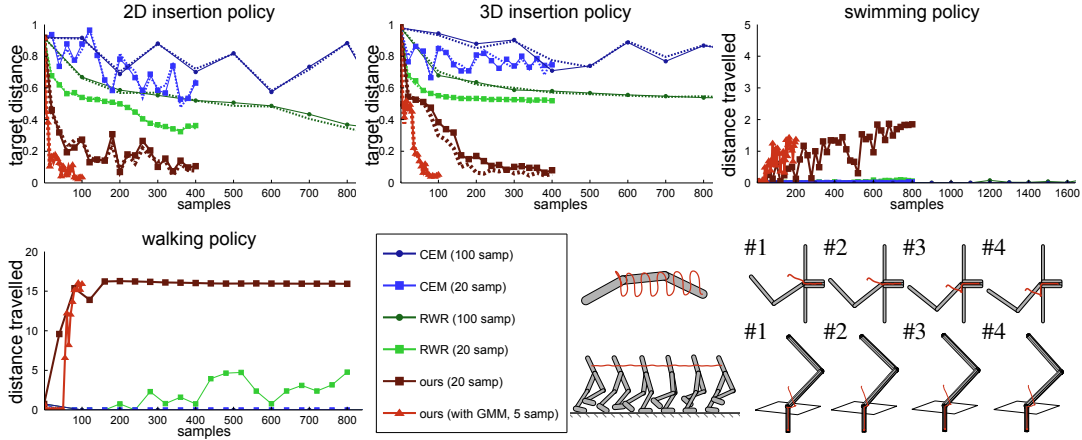

Figure 2: Comparison on neural network policies. For insertion, the policy was trained to search for an unknown slot position on four slot positions (shown above), and generalization to new positions is graphed with dashed lines. Note how the end effector (in red) follows the surface to find the slot, and how the swimming gait is smoother due to the stationary policy (also see supplementary video).

to weighted samples, including PoWER and PI2 [11, 24, 22]. PILCO is a model-based method that uses a Gaussian process to learn a global dynamics model that is used to optimize the policy. REPS and PILCO require solving large nonlinear optimizations at each iteration, while our method does not. Our method used 5 rollouts with the GMM, and 20 without. Due to its computational cost, PILCO was provided with 5 rollouts per iteration, while other prior methods used 20 and 100.

Our method learned much more effective controllers with fewer samples, especially when using the GMM. On 3D insertion, it outperformed the iLQG baseline, which used a known model. Contact discontinuities cause problems for derivative-based methods like iLQG, as well as methods like PILCO that learn a smooth global dynamics model. We use a time-varying local model, which preserves more detail, and fitting the model to samples has a smoothing effect that mitigates discontinuity issues. Prior policy search methods could servo to the hole, but were unable to insert the peg. On the octopus arm, our method succeeded despite the high dimensionality of the state and action spaces.[1] Prior work used simplified "macro-actions" to solve this task, while our method directly controlled each degree of freedom [6]. Our method also successfully learned a swimming gait, while prior model-free methods could not initiate forward motion.[2] PILCO also learned an effective gait due to the smooth dynamics of this task, but its GP-based optimization required orders of magnitude more computation time than our method, taking about 50 minutes per iteration.

These results suggest that our method combines the sample efficiency of model-based methods with the versatility of model-free techniques. However, this method is designed specifically for linear-Gaussian controllers. In the next section, we present results for learning more general policies with our method, using the linear-Gaussian controllers within the framework of guided policy search.

## 5.2 Neural Network Policy Learning with Guided Policy Search

By using our method with guided policy search, we can learn arbitrary parameterized policies. Figure 2 shows results for training neural network policies for each task, with comparisons to prior methods that optimize the policy parameters directly.[3] On swimming, our method achieved similar performance to the linear-Gaussian case, but since the neural network policy was stationary, the resulting gait was much smoother. Previous methods could only solve this task with 100 samples per iteration, with RWR eventually obtaining a distance of 0.5m after 4000 samples, and CEM reaching 2.1m after 3000. Our method was able to reach such distances with many fewer samples.

Generating walking from scratch is extremely challenging even with a known model. We therefore initialize the gait from demonstration, as in prior work [12]. The supplementary website also shows some gaits generated from scratch. To generate the initial samples, we assume that the demonstration can be stabilized with a linear feedback controller. Building such controllers around examples has been addressed in prior work [3]. The RWR and CEM policies were initialized with samples from this controller to provide a fair comparison. The walker used 5 samples per iteration with the GMM, and 40 without it. The graph shows the average distance travelled on rollouts that did not fall, and shows that only our method was able to learn walking policies that succeeded consistently.

On the insertion tasks, the neural network was trained to insert the peg without precise knowledge of the position of the hole, making this a partially observed problem. The holes were placed in a region of radius 0.2 units in 2D and 0.1 units in 3D. The policies were trained on four different hole positions, and then tested on four new hole positions to evaluate generalization. The generalization results are shown with dashed lines in Figure 2. The position of the hole was not provided to the neural network, and the policies therefore had to find the hole by "feeling" for it, with only joint angles and velocities as input. Only our method could acquire a successful strategy to locate both the training and test holes, although RWR was eventually able to insert the peg into one of the four holes in 2D. This task illustrates one of the advantages of learning expressive neural network policies, since no single trajectory-based policy can represent such a search strategy. Videos of the learned policies can be viewed at `http://rll.berkeley.edu/nips2014gps/`.

## 6 Discussion

We presented an algorithm that can optimize linear-Gaussian controllers under unknown dynamics by iteratively fitting local linear dynamics models, with a background dynamics distribution acting as a prior to reduce the sample complexity. We showed that this approach can be used to train arbitrary parameterized policies within the framework of guided policy search, where the parameterized policy is optimized to match the linear-Gaussian controllers. In our evaluation, we show that this method can train complex neural network policies that act intelligently in partially observed environments, even for tasks that cannot be solved with direct model-free policy search.

By using local linear models, our method is able to outperform model-free policy search methods. On the other hand, the learned models are highly local and time-varying, in contrast to model-based methods that rely on learning an effective global model [4]. This allows our method to handle even the complicated and discontinuous dynamics encountered in the peg insertion task, which we show present a challenge for model-based methods that use smooth dynamics models [5]. Our approach occupies a middle group between model-based and model-free techniques, allowing it to learn rapidly, while still succeeding in domains where the true model is difficult to learn.

Our use of a KL-divergence constraint during trajectory optimization parallels several prior model-free methods [1, 19, 17, 20, 16]. Trajectory-centric policy learning has also been explored in detail in robotics, with a focus on dynamic movement primitives (DMPs) [8, 24]. Time-varying linear-Gaussian controllers are in general more expressive, though they incorporate less prior information. DMPs constrain the final state to a goal state, and only encode target states, relying on an existing controller to track those states with suitable controls.

The improved performance of our method is due in part to the use of stronger assumptions about the task, compared to general policy search methods. For instance, we assume that time-varying linear-Gaussians are a reasonable local approximation for the dynamics. While this assumption is sensible for physical systems, it would require additional work to extend to hybrid discrete-continuous tasks.

Our method also suggests some promising future directions. Since the parameterized policy is trained directly on samples from the real world, it can incorporate sensory information that is difficult to simulate but useful in partially observed domains, such as force sensors on a robotic gripper, or even camera images, while the linear-Gaussian controllers are trained directly on the true state under known, controlled conditions, as in our peg insertion experiments. This could provide for superior generalization for partially observed tasks that are otherwise extremely challenging to learn.

**Acknowledgments**

This research was partly funded by a DARPA Young Faculty Award #D13AP0046.

## Footnotes

[1]The high dimensionality of the octopus arm made it difficult to run PILCO, though in principle, such methods should perform well on this task given the arm's smooth dynamics.

[2]Even iLQG requires many iterations to initiate any forward motion, but then makes rapid progress. This suggests that prior methods were simply unable to get over the initial threshold of initiating forward movement.

[3]PILCO cannot optimize neural network controllers, and we could not obtain reasonable results with REPS. Prior applications of REPS generally focus on simpler, lower-dimensional policy classes [17, 16].

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
