[Supplementary Material]

# A  Dynamic Programming with Guided Policy Search

In this appendix, we present the details of the dynamic programming algorithm for optimizing the trajectory distributions in the presence of policy KL-divergence terms. Recall that the trajectory optimization objective is

$$\mathcal{L}_{\text{GPS}}(p) = E_{p(\tau)}[\ell(\tau)] + \sum_{t=1}^{T} \lambda_t D_{\text{KL}}(p(\mathbf{x}_t)\pi_\theta(\mathbf{u}_t|\mathbf{x}_t)\|p(\mathbf{x}_t,\mathbf{u}_t)).$$

Adding the KL-divergence constraint against the previous trajectory distribution $\hat{p}(\tau)$ with Lagrange multiplier $\eta$, the goal of the dynamic programming pass is to minimize the following objective:

$$\mathcal{L}_{\text{GPS}}(p) = E_{p(\tau)}[\ell(\tau) - \eta\log\hat{p}(\tau)] - \eta\mathcal{H}(p) + \sum_{t=1}^{T} \lambda_t D_{\text{KL}}(p(\mathbf{x}_t)\pi_\theta(\mathbf{u}_t|\mathbf{x}_t)\|p(\mathbf{x}_t,\mathbf{u}_t)).$$

Dividing by $\eta$ and letting $\tilde{\ell}(\mathbf{x}_t,\mathbf{u}_t) = \frac{1}{\eta}\ell(\mathbf{x}_t,\mathbf{u}_t) - \log\hat{p}(\mathbf{u}_t|\mathbf{x}_t)$, we can simplify the objective to be

$$\frac{1}{\eta}\mathcal{L}_{\text{GPS}}(p) = E_{p(\tau)}[\tilde{\ell}(\tau)] - \mathcal{H}(p) + \sum_{t=1}^{T} \frac{\lambda_t}{\eta} D_{\text{KL}}(p(\mathbf{x}_t)\pi_\theta(\mathbf{u}_t|\mathbf{x}_t)\|p(\mathbf{x}_t,\mathbf{u}_t)).$$

The distribution $p(\tau)$ factorizes over time steps into the linear-Gaussian dynamics terms and the conditional action terms $p(\mathbf{u}_t|\mathbf{x}_t) = \mathcal{N}(\mathbf{K}_t\mathbf{x}_t + \mathbf{k}_t, \mathbf{C}_t)$. We will use $(\hat{\mathbf{x}}_t, \hat{\mathbf{u}}_t)^{\text{T}}$ to denote the mean of the marginals $p(\mathbf{x}_t,\mathbf{u}_t)$ and $\Sigma_t$ to denote their covariance, while $\mathbf{S}_t$ will denote the covariance of $p(\mathbf{x}_t)$. Using a second order Taylor expansion of the cost and a linear-Gaussian approximation to the policy $\pi_\theta(\mathbf{u}_t|\mathbf{x}_t)$, the objective $\mathcal{L}_{\text{GPS}}(p)$ becomes

$$\mathcal{L}_{\text{GPS}}(p) \approx \sum_{t=1}^{T} \frac{1}{2}\begin{bmatrix}\hat{\mathbf{x}}_t\\\hat{\mathbf{u}}_t\end{bmatrix}^{\text{T}}\tilde{\ell}_{\mathbf{xu},\mathbf{xu}t}\begin{bmatrix}\hat{\mathbf{x}}_t\\\hat{\mathbf{u}}_t\end{bmatrix} + \begin{bmatrix}\hat{\mathbf{x}}_t\\\hat{\mathbf{u}}_t\end{bmatrix}^{\text{T}}\tilde{\ell}_{\mathbf{xu}t} + \frac{1}{2}\text{tr}\left(\Sigma_t\tilde{\ell}_{\mathbf{xu},\mathbf{xu}t}\right) - \frac{1}{2}\log|\mathbf{C}_t| +$$

$$\frac{\lambda_t}{2\eta}\log|\mathbf{C}_t| + \frac{\lambda_t}{2\eta}(\hat{\mathbf{u}}_t - \mu_t^\pi(\hat{\mathbf{x}}_t))^{\text{T}}\mathbf{C}_t^{-1}(\hat{\mathbf{u}}_t - \mu_t^\pi(\hat{\mathbf{x}}_t)) + \frac{\lambda_t}{2\eta}\text{tr}\left(\mathbf{C}_t^{-1}\Sigma_t^\pi\right) +$$

$$\frac{\lambda_t}{2\eta}\text{tr}\left(\mathbf{S}_t\left(\mathbf{K}_t - \mu_{\mathbf{x}t}^\pi(\hat{\mathbf{x}}_t)\right)^{\text{T}}\mathbf{C}_t^{-1}\left(\mathbf{K}_t - \mu_{\mathbf{x}t}^\pi(\hat{\mathbf{x}}_t)\right)\right),$$

where $\pi_\theta(\mathbf{u}_t|\mathbf{x}_t) \approx \mathcal{N}(\mathbf{u}_t; \mu_{\mathbf{x}t}^\pi(\hat{\mathbf{x}}_t)\mathbf{x}_t + \mu_t^\pi(\hat{\mathbf{x}}_t), \Sigma_t^\pi)$ is the linear-Gaussian approximation to the policy around $(\hat{\mathbf{x}}_t, \hat{\mathbf{u}}_t)$. This linearization is fitted to the samples in the same fashion as the dynamics. In fact, the same type of GMM can be used to enable this fitting to be done with a small number of samples. Once this objective is formed, it can be optimized with a backward dynamic programming pass analogous to iLQG, which is described in previous work [14], to produce updated linear-Gaussian parameters $\hat{\mathbf{x}}_t$, $\hat{\mathbf{u}}_t$, $\mathbf{K}_t$, and $\mathbf{C}_t$.

After performing the dynamic programming pass and updating $\hat{\mathbf{x}}_t$, $\hat{\mathbf{u}}_t$, $\mathbf{K}_t$, and $\mathbf{C}_t$, the policy parameters $\theta$ are optimized to minimize the KL-divergence $D_{\text{KL}}(p(\mathbf{x}_t)\pi_\theta(\mathbf{u}_t|\mathbf{x}_t)\|p(\mathbf{x}_t,\mathbf{u}_t))$, weighted by $\lambda_t$ at each time step. This is equivalent to minimizing $E_{p(\mathbf{x}_t)}[D_{\text{KL}}(\pi_\theta(\mathbf{u}_t|\mathbf{x}_t)\|p(\mathbf{x}_t,\mathbf{u}_t))]$, and we can use the same samples to estimate this expectation that we will use to fit the new dynamics corresponding to the updated trajectory distribution, as shown in Algorithm 1. Note that we must sample before updating the policy, so that the samples are drawn from the new $p(\tau)$, optimize the policy, and only then update $p(\tau)$ using the new samples. Rewriting the policy objective as a sample average, we get

$$\mathcal{L}(\theta) = \sum_{t=1}^{T} \lambda_t \sum_{i=1}^{N} \frac{1}{2}\left\{\text{tr}(\Sigma_t^\pi(\mathbf{x}_{ti})\mathbf{C}_t^{-1}) - \log|\Sigma^\pi(\mathbf{x}_{ti})| +\right.$$

$$\left.(\mathbf{K}_t\mathbf{x}_{ti}+\mathbf{k}_t-\mu^\pi(\mathbf{x}_{ti}))^{\text{T}}\mathbf{C}_t^{-1}(\mathbf{K}_t\mathbf{x}_{ti}+\mathbf{k}_t-\mu^\pi(\mathbf{x}_{ti}))\right\},$$

where we assume the policy is conditional Gaussian with the mean $\mu^\pi(\mathbf{x}_t)$ and covariance $\Sigma_t^\pi(\mathbf{x}_t)$ any function of the state $\mathbf{x}_t$. Note that the objective is a least squares objective on the policy mean, weighted by the conditional covariance of the trajectory, and can be minimized with any unconstrained nonlinear optimization algorithm. We use LBFGS in our implementation. When using the GMM and a small number of samples per iteration, we augment the training set with samples from previous iterations. Although this is not strictly correct, since the samples no longer all come from $p(\mathbf{x}_t)$, we still use the actions from the new trajectory distribution, and this tends to produce significantly better results in practice.

# B Simulation Parameters and Cost Functions

In this appendix, we present the physical parameters of each system in our experimental evaluation, as well as the cost function used for each task. All of the systems were simulated using a rigid body simulation package, with simulated frictional contacts and torque motors at the joints used for actuation.

**Peg insertion:** The 2D peg insertion task has 6 state dimensions (joint angles and angular velocities) and 2 action dimensions. The 3D version of the task has 12 state dimensions, since the arm has 3 degrees of freedom at the shoulder, 1 at the elbow, and 2 at the wrist. Trials were 8 seconds in length and simulated at 100 Hz, resulting in 800 time steps per rollout. The cost function is given by

$$\ell(\mathbf{x}_t, \mathbf{u}_t) = \frac{1}{2}w_{\mathbf{u}}\|\mathbf{u}_t\|^2 + w_{\mathbf{p}}\ell_{12}(\mathbf{p}_{\mathbf{x}_t} - \mathbf{p}^\star),$$

where $\mathbf{p}_{\mathbf{x}_t}$ is the position of the end effector for state $\mathbf{x}_t$, $\mathbf{p}^\star$ is the desired end effector position at the bottom of the slot, and the norm $\ell_{12}(z)$ is given by $\frac{1}{2}\|z\|^2 + \sqrt{\alpha + z^2}$, which corresponds to the sum of an $\ell_2$ and soft $\ell_1$ norm. We use this norm to encourage the peg to precisely reach the target position at the bottom of the hole, but to also receive a larger penalty when far away. The task also works well in 2D with a simple $\ell_2$ penalty, though we found that the 3D version of the task takes longer to insert the peg all the way into the hole without the $\ell_1$-like square root term. The weights were set to $w_{\mathbf{u}} = 10^{-6}$ and $w_{\mathbf{p}} = 1$.

**Octopus arm:** The octopus arm consists of six four-sided chambers. Each edge of each chamber is a simulated muscle, and actions correspond to contracting or relaxing the muscle. The state space consists of the positions and velocities of the chamber vertices. The midpoint of one edge of the first chamber is fixed, resulting in a total of 25 degrees of freedom: the 2D positions of the 12 unconstrained points, and the orientation of the first edge. Including velocities, the total dimensionality of the state space is 50. The cost function depends on the activation of the muscles and distance between the tip of the arm and the target point, in the same way as for peg insertion. The weights are set to $w_{\mathbf{u}} = 10^{-3}$ and $w_{\mathbf{p}} = 1$.

**Swimming:** The swimmer consists of 3 links and 5 degrees of freedom, including the global position and orientation which, together with the velocities, produces a 10 dimensional state space. The swimmer has 2 action dimensions corresponding to the torques between joints. The simulation applied drag on each link of the swimmer to roughly simulate a fluid, allowing it to propel itself. The rollouts were 20 seconds in length at 20 Hz, resulting in 400 time steps per rollout. The cost function for the swimmer is given by

$$\ell(\mathbf{x}_t, \mathbf{u}_t) = \frac{1}{2}w_{\mathbf{u}}\|\mathbf{u}_t\|^t + \frac{1}{2}w_v\|v_{x\mathbf{x}_t} - v_x^\star\|^2$$

where $v_{x\mathbf{x}_t}$ is the horizontal velocity, $v_x^\star = 2.0$m/s, and the weights were $w_{\mathbf{u}} = 2\cdot10^{-5}$ and $w_v = 1$.

**Walking:** The bipedal walker consists of a torso and two legs, each with three links, for a total of 9 degrees of freedom and 18 dimensions, with velocity, and 6 action dimensions. The simulation ran for 5 seconds at 100 Hz, for a total of 500 time steps. The cost function is given by

$$\ell(\mathbf{x}_t, \mathbf{u}_t) = \frac{1}{2}w_{\mathbf{u}}\|\mathbf{u}_t\|^t + \frac{1}{2}w_v\|v_{x\mathbf{x}_t} - v_x^\star\|^2 + \frac{1}{2}w_h\|p_{y\mathbf{x}_t} - p_y^\star\|^2$$

where $v_{x\mathbf{x}_t}$ is again the horizontal velocity, $p_{y\mathbf{x}_t}$ is the vertical position of the root, $v_x^\star = 2.1$m/s, $p_y^\star = 1.1$m, and the weights were set to $w_{\mathbf{u}} = 10^{-4}$, $w_v = 1$, and $w_h = 1$.