[Reviews · NeurIPS 2014]

Submitted by Assigned_Reviewer_12

Summary:
The paper presents a sample-efficient policy search algorithm for large, continuous reinforcement learning problems. In contrast to existing model-based policy search algorithms, the approach presented in this paper tries to learn local models in form of linear Gaussian controllers. Given the information (rollouts) from these linear local models, a global, nonlinear policy can then be learned using an arbitrary parametrization scheme. The so-called Guided Policy Search approach alternates between (local) trajectory optimization and (global) policy search in an iterative fashion. In their experiments, the authors show that the approach outperforms various state-of-the-art Policy Search methods, e.g., REPS, PILCO etc. Experiments where conducted in (mostly 2D) dynamics simulations involving the continuous control of multi-linked agents.

Quality:
This is a well-written paper that addresses a topic which is relevant to the NIPS. Policy search algorithms have gained significant popularity in recent years and the work presented in this paper adds various important and interesting insights. Training a global model from locally linear models in a supervised fashion is an excellent idea! The results presented in the experiments section show that the approach can outperform well-known PS methods. However, it is at times difficult to judge the quality of the method since the proposed experiments are new and (to the best of my knowledge) have not been used by other researchers. Deisenroth et al. showed that PILCO can learn the "Cart-Pole Swing-up" task within a few trials. Can GPS do the same? I also wonder whether the local models can influence each other during dynamics fitting. For this not to happen, you need to ensure that the rollouts are only generated in the timeframe in which a particular local model is the expert. After reading the paper several times, it was still unclear to me how this is achieved. Especially when training Neural Networks, conflicts between several local datasets can have an extremely negative effect. That being said, I really like the fact that different types of policy representations can be used. While they have been out of fashion for controls, this paper clearly shows that (trained the right way) neural networks can we powerful tools for continuous problems.
Clarity:
All sections of the paper are clear and understandable. The authors did a very good job explaining the reasons behind the each choice made in the development of the algorithm. I particularly like that information about practical/empirical choices (e.g. parameter values, how to speed up learning) is shared with the reader.
Originality:
The paper bears several similarities with well-known PS algorithms, in particular REPS. The use of KL divergence when learning policies was made popular by the REPS algorithm. Using supervised learning bears some resemblance to model-based RL. However, I think that these similarities are mostly superficial. At its core, the GPS approach differs significantly from exisiting PS approaches. In constract to model-based RL, the approach here learns the Policy in a supervised learning fashion rather than a model of the environment. Using local models has various beneficial effects, e.g. simpler training, stability of the system, convergence etc.
Significance:
I think that the paper makes a significant and valuable contribution to the Policy Search and Reinforcement Learning community. While the approach builds upon similar, existing methods, it has various interesting new ideas. I recommend the publication of this paper.
Summary: This is a well-written paper that introduces a method for sample-efficient learning of policies in unknown environments. Turning policy search into a supervised learning problem is a very interesting new research direction and will hopefully lead to more insights into this problem.

Submitted by Assigned_Reviewer_20

Summary of the paper:
The paper proposes a trajectory optimization method derived from iLQG, where instead of using the linearization of a known model, the linearized dynamics is acquired from samples. In order to ensure proper convergence in iLQG a backtracking line search can be implemented in the forward pass to ensure that the new trajectory does not differ too much from the previous one. In this paper, since the dynamics is unknown the line search is replaced by the addition of a constraint on the KL divergence between the previous and the new trajectory distribution during optimization to constrain the change in the new trajectory. A GMM model is used as a prior for the sampled unknown dynamics. Moreover, a parametrized policy is learned through guided policy search using the trajectory optimization framework with a neural network to approximate the policy. Experiments on 3 different simplified robotics problem as well as comparisons with other methods are provided to support the claims of the paper.

Comments:
The paper is interesting and well-written and in general it addresses an important topic for robotics research. In terms of the novelty of the approach, the main contribution of the paper is to show how iterative LQG can be used in a model-free context while ensuring convergence by replacing the backtracking line-search with a KL divergence constraint. While this is an interesting approach, one could argue that this technical improvement is rather incremental given knowledge on iLQG methods and linear Gaussian approximation of the dynamics. The guided policy search approach is also interesting but is also related to previous work, for example the work of Mordatch et al. (RSS 2014). One good aspect of the experiments is that it nicely shows that the proposed approach converges faster than other methods, even without the use of GMM as a prior model.

One comment with the use of a GMM prior is that this prior somehow already constitutes a model and therefore it is expected that the proposed approach performs close to model-based trajectory optimization methods. Indeed, it is mentioned in 3.2 that large mixtures that modeled the dynamics with high details produced the best results. This is to be expected since this basically already provide a good dynamic model. One question that arises is how is this GMM constructed? (i.e. what is the method to sample the potentially high dimensional space while ensuring safety on a real robot?). Additional comments on the likelihood that such an approach could scale to more realistic robot models where such GMM construction would be extremely challenging would also be useful.

Another possible important limitation of the approach is that it already requires an example demonstration for the walking task, which is only a 2D walking task. It is well-known that 3D walking is much more difficult than 2D walking (where relatively simple feedback control solutions are known to exist for this type of walking, e.g. the work of M. Spong or R. Gregg or J. Grizzle on the topic). Having an example demonstration means that it is already possible to solve the problem in some sense. It would be interesting that the paper comments on this issue. It also raises the question on how this method can really scale to more complicated problems (e.g. 3D walking or swimming with more than 2 joints). Indeed, assuming that the GMM prior is precise, it is surprising that the optimizer does not find a walking solution since the iLQG method can find such solutions with 3D walking and many more DOFs.

Another comment concerns the comparison with iLQG for the tasks involving contacts. What is the contact model used in this evaluations and what is the contact model used by the iLQG algorithm? Tassa and colleagues have shown very good results with tasks involving contacts and many DOFs using iLQG, one of the reason residing in the choice of a contact model that is more “friendly” with iLQG optimization techniques. For a fair comparison, the paper should compare the tasks using this contact model for iLQG.
Summary: Overall the paper is interesting and well written and the comparison with other learning approaches is very useful. My main concerns are related to 1) the use of KL divergence for line search which is a rather incremental change of iLQG and 2) scalability issues of the approach for more realistic scenarios.

Submitted by Assigned_Reviewer_22

Summary:

The authors propose the combined use of trajectory optimization and policy learning for systems with unknown dynamics approximated by linear local models. The proposed trajectory optimization algorithm is based on DDP/iLQG methods. The local linear models are assumed Gaussian and obtained from samples returned by the previous policy solution. The paper introduces the iLQG-based method such that the approximated local models can be used. Optimization with approximated models is made possible by limiting the changes of the modified iLQG solution using KL-divergence as a constraint. The paper also presents the use of their trajectory optimization method within the framework of Guided Policy Search previously proposed in [11] such that richer parameterized policies can be learned by iteratively swapping between the trajectory optimization and the policy learning steps. Results are evaluated in simulation.

Quality:

The paper is of high quality. It is clearly written. Sometimes the paper relies too much on prose (see comments below) and support for a few statements are not clearly found. The paper is of high interest for the robotics and robot learning community especially due to the "unknown dynamics" feature, however, large part of the merit is attributed to the fact that it builds upon the the contributions of [11, 12, 13]. The evaluation of the method is of high quality and extensive and several relevant state-of-the-art methods are selected for comparison. The simulated tasks are interesting and show in a clear way the differences in performance.

Clarity and Questions:

Certain parts of the paper are not clear and although he supporting information may be somewhere in the paper it is not always easy to find.
. Please, define the use of "unknown dynamics" and "hybrid" right at the introduction. Unknown dynamics inevitably induces the "model-free" image, but the paper does not seem to consider itself as a model-free method. At the same time, the idea of creating a model from samples is usually regarded as model-based (such as PILCO). I personally would not introduce the new term "unknown dynamics" but rather use something like "local Gaussian models" instead.
. In line 52, "parameterized policy never needs to be executed on the real system" needs clarification. What I understand from this sentence is that you would use the Gaussian controller during the whole learning process, and only at the end learn the parameterized policy based on what the Gaussian controller found (?). In that case it seems that you are not using the GPS framework as the parameterized policy is not being influenced by the trajectory optimization and vice-versa.
. Line 59, the conclusion "...transforms a discontinuous deterministic problem into a smooth stochastic one" is very strong but the intuition that lead to it is not convincing. Is there a better way to explain it? A related request: since simulations with contact are difficult to model and local models do not capture discontinuity the paper would become much stronger with experimental results on a tasks with real physical contact or impact.
. Although the GMM for background dynamics distribution seems to be very beneficial (in lines 400-401 it even seems to be intrinsic part of the method), it is only in Section 3.2 that GMMs are firstly mentioned. Until this point the whole method is explained as if there is no GMM. This can be a little confusing as at this point I do not know if GMM is part of the method or GMM is an extra feature that improves the method. As it seems to be always a good thing, why not present the method as unknown dynamics approximated by a mixture of Gaussians?
. How does the system behave under saturation? Any real physical robot will have actuation limits. Does it compromise the induction of Gaussian trajectory distributions if the commands from the controller are saturated?
. The fact that the iLQG with a true model performs worse than your method (which builds on iLQG but uses approximated linear models) is counter-intuitive. How it is possible?

Originality and Significance:

The paper is incremental in relation to the works of [11,12,13] although the contribution it adds is significant. The idea of being able to optimize trajectories and design a controller for an unknown system dynamics is of high interest for the robotics community, which was not previously addressed by [11,12,13].

Summary: This paper is, in general, well written and easy to understand. It addresses the ability to apply the GPS framework [11,12,13] on systems with unknown dynamics by creating local Gaussian models. Although the paper is incremental in relation to existing work, the whole idea is of high interest for the robotics community.
Author Feedback
Author rebuttal: We thank the reviewers for their positive feedback and constructive criticism. We clarify some of the issues raised in the reviews below (these clarifications will also be added in the final version):

Several reviewers asked about the comparison to iLQG. We emphasize that the comparison is mainly intended to show that our method can handle contact discontinuities, which typically pose a major challenge for DDP-like methods (this is why iLQG was unable to solve 3D insertion). As reviewer #20 pointed out, techniques like soft contacts can be used to make the problem easier for iLQG; the purpose of the comparison was to show that our method could still handle the task even with discontinuities (which are present when sampling from the real dynamics of a physical system). We will clarify this point.

Reviewer #20:

In regard to how the GMM is constructed, whether it could serve as a global model, and how well it would scale to larger systems: No additional samples are taken to construct the GMM, it is built entirely out of samples that are already collected along rollouts of the current and previous linear Gaussian controllers. The GMM can serve as a global model for simple tasks, but in general, it does not need to be a good *forward* model, since it is only used as a prior. As discussed in 3.2, it may be quite a poor forward model when the contact boundary is poorly approximated by the boundary between two Gaussians. Because the GMM does not need to serve as a good global model, there is much less pressure on it to successfully model high dimensional dynamics. We showed results for problems of up to 18 dimensions, which is already sufficient for many robotic systems, such as 7 DoF arms, on which we are currently applying our method. In the final version, we will also present results for a 100-dimensional octopus arm task, which our method is also able to learn in about 20 iterations with 5 samples per iteration.

In regard to walking, we were able to get comparable locomotion from scratch using both our method and iLQG -- the walker made good forward progress, but did not appear naturalistic. Prior iLQG locomotion results employed shaped rewards and/or careful initialization (see e.g. Erez et al "An integrated system for real-time...") to produce good walking behaviors, so it's likely that more careful reward shaping would produce good locomotion results without examples. We agree that full 3D walking will be more challenging (we have not yet tested it), though likely outside of the scope of this work.

Reviewer #22:

In regard to actuation limits, we have recently tested the method under actuation limits, both in simulation and on a physical robot. The results are preliminary, but the method does not appear to perform worse under actuation limits (which are modeled as part of the dynamics -- i.e. the actions are unbounded, but have no additional effect beyond a certain point).

We will clarify our use of the term "hybrid," which we use to illustrate that our method combines properties of both model-free and model-based methods, as the reviewer noted.

In regard to "transforming a discontinuous problem into a smooth stochastic one," this statement applies equally to any stochastic policy search procedure, where discontinuous dynamics are averaged over multiple samples (as opposed to deterministic DDP-like methods), though perhaps this is not phrased very clearly. We will revise it to clarify this.

In regard to line 52, the parameterized policy influences the trajectory optimization (and vice versa), but all rollouts on the real system use the linear Gaussian controller. This is consistent with the GPS framework (for instance, see [12] Section 8 paragraph 2 for a discussion).

Reviewer #12:

In regard to the experiments, the swimmer is a fairly common benchmark (see Coulom, Todorov, etc). The other tasks were chosen to present a major challenge both for our method and prior approaches. In particular, we are not aware of standard benchmarks that involve hard contacts. However, we do hope to add more standard benchmarks to the final version, such as the octopus arm and/or underactuated swing-up (subject to space constraints).

It is quite likely that PILCO would outperform our method on simple tasks with smooth dynamics, where the GP model has the right inductive bias. For example, the swimmer already shows PILCO performing well. The advantage of our approach is that it can extend to a broader range of tasks, including contact-rich tasks that are very common in robotics and difficult to model with smooth GPs. That said, the GP model learned by PILCO could serve as a very good prior (instead of the GMM).

In regard to sharing of information: we're uncertain whether the reviewer is referring to dynamics or policy learning. In the case of dynamics learning, information is shared between the locally linear models by means of the background dynamics distribution (GMM), which incorporates data from all time steps. In the case of policy learning, agreement between different trajectories is enforced by a (soft) constraint that each trajectory distribution should agree with the neural network policy, which gradually brings the policy and all trajectory distributions into agreement.

We will further discuss the connection to REPS and other algorithms in more detail in the final version (perhaps in an appendix if space constraints are too tight).